# Learning with Language Inference and Tips for Continual Reinforcement Learning

## Abstract

Acquiring a generalizable policy by continually learning a sequence of tasks is a natural human skill yet challenging for current reinforcement learning algorithms. This is largely due to the dilemma that the agent is expected to quickly adapt to new tasks (plasticity) while retaining the common knowledge from previous tasks (stability). In this work, we present a scheme referred to as "Learning with Language Inference and Tips (LLIT)", which introduces a rewarding mechanism to parse and ground human knowledge in natural language form to the task space and produces an interpretable policy for each task in task-agnostic setting. LLIT trains a shared policy for each task by inferring and embedding the tips and content of the task. The language instructions inferred by the large language model (LLM) are then used to pre-train an auxiliary reward model with observations' embedding, thereby extracting the semantic representations in tasks. Simultaneously, the instructions and tips embedding will be collected and organized as a prompt pool to capture the correlation among tasks. Hence, closely related tasks exhibit greater neuron overlap in the policy network, stemming from shared semantics, which effectively curbs cross-task interference and forgetfulness. Given the auxiliary reward model trained on previous tasks that interprets human knowledge in natural language, new task adaptation reduces to highly efficient tips aggregation and sub-network finetuning. In experimental studies, LLIT achieves a desirable plasticity-stability trade-off without any task-specfic information. It also outperforms existing continual RL methods in terms of overall performance, forgetting reduction, and adaptation to unseen tasks. Our code is available at https://github.com/llm4crl/LLIT.

## 1 Introduction

Reinforcement learning(RL) has demonstrated remarkable performance and great potential on learning individual tasks, such as playing strategic games(Silver et al., 2016; Vinyals et al., 2019), robotic control (Kober et al., 2013; Kormushev et al., 2013; Polydoros & Nalpantidis, 2017) and autonomous driving(Aradi, 2020; Kiran et al., 2021). However, it is hard for RL to perform well when learning a stream of tasks sequentially, due to catastrophic forgetting and difficulty to transfer knowledge between tasks(Bengio et al., 2020; Khetarpal et al., 2022). Consequently, training a single policy that can properly handle to all learned tasks or even swiftly adapt and generalize to unseen ones still remains as a major challenge. This problem is commonly referred to continual or lifelong RL(Mendez et al., 2020) and has attracted growing interest in the RL community.

The plasticity-stability trade-off(Khetarpal et al., 2022) is a fundamental and enduring issue in continual RL.The RL policy should concurrently preserve and utilize task-related knowledge in past(stability) while remaining adaptive to novel tasks without being interfered by previous tasks(plasticity). In practice, this issue is a key factor in improving the efficiency of continual RL and the generalization capability of its learned policy. Recently, due to the thriving development of large language model (LLM), instructions in natural language derived from a pre-trained LLM can efficiently guide the policy with human knowledge, mitigating the necessity of experience replay and the associated memory and computational overhead. Moreover, The instructions obtained by inference can not only optimize the policy gradient updates effectively, but also serve as the auxiliary representation to transfer the shared skills positively across tasks. Hence, the stability can be signif-

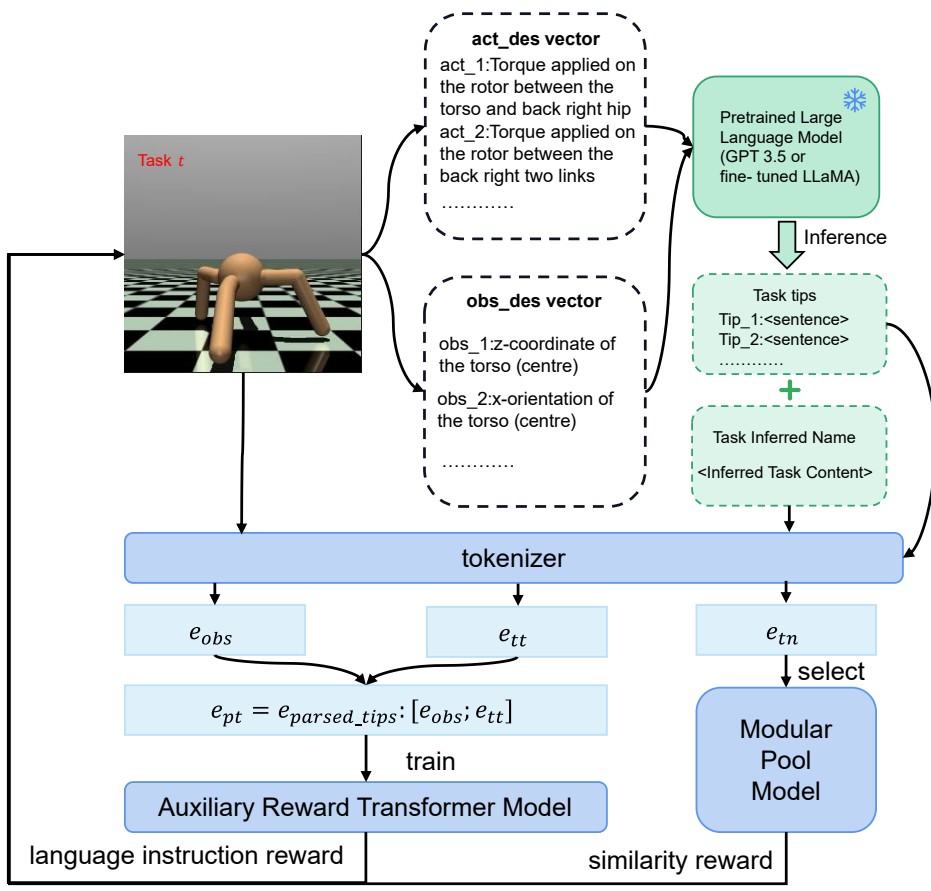

Figure 1: **The workflow and key components of LLIT.**

icantly enhanced as the number of tasks increases with better plasticity achieving faster adaptation and better generalization to new tasks.

To directly address the plasticity-stability dilemma and overcome the drawbacks of previous work, we choose to incorporate knowledge in the form of natural language into continual RL. Then, given only a textual description of the observation space and the action space of a previously learned or unseen task, a set of inferred tips to solve the task and inferred task content can be automatically and efficiently extracted from the large language model. This aligns with the concept of prompting in recent large language models but distinguishes from existing methods, which either optimize a task-specific policy from scratch(Boschini et al., 2022) or select a policy from the pool of pre-trained models for each task(Kessler et al., 2022). To this end, we propose to learn a reward model along with a prompt pool to encode the knowledge as an auxiliary signal, which provide high-level semantic understandings for the policy, which also extract cross-tasks skills from the semantics to enable the continual learning. We call this approach "Learning with Language Inference and Tips (LLIT)".

In Fig.1, given the t-th task $t$, the task tips and content are generated by a frozen and pre-trained LLM, which are tokenized along with observations into three embeddings $e_{tt}$, $e_{tn}$ and $e_{obs}$ separately. Next, $e_{tt}$ and $e_{obs}$ are element-wise concatenated as input to a transformer model, which is trained as an auxiliary reward generator to produce the language instruction reward. Meanwhile, the embedding of the inferred task content $e_{tn}$ is put into a prompt pool, which exploits the task correlations in both the embedding and prompt spaces. This leads to efficient usage of the semantic interpretations and goal-oriented optimization of the trade-off between plasticity and stability in that relevant tasks can reuse skills by sharing more tips and parameters (good plasticity and fast adaptation) while the harmful interference between irrelevant tasks can be largely avoided by sharing less

or no tips and parameters (high stability and less forgetting). Moreover, due to the prompt pool, it is unnecessary for to store any experiences of previous tasks in replay buffer, requiring much less computation and memory than rehearsal-based methodsSerra et al. (2018); Rolnick et al. (2019). Furthermore, the prompt pool in LLIT, as an efficient task adaptation method, can extract policy sub-networks for unseen tasks and thus leads to a more generalizable meta-policy.

In our work, we show that on Continual World benchmarks and randomly sampled task sequences that consist of tasks from different domains, LLIT outperforms overall performance most baselines in terms of overall performance, forgetting reduction, and adaption to unseen tasks (Table 1).We also make a comprehensive ablation study (Table 2) and confirm the importance of inferring language instructions and obtaining prompt pool. In addition, our empirical analysis shows that auxiliary reward model trained on language instructions converges fast and can judge the value of current trajectory from the aspect of semantics precisely(Table 2), while the learned prompt pool captures the semantic correlations among tasks(Fig.**??**).

## 2 PRELIMINARIES AND RELATED WORK

We follow the task-incremental setting in previous work(Wolczyk et al., 2021; 2022; Khetarpal et al., 2022; Rolnick et al., 2019; Mendez et al., 2020; Chaudhry et al., 2018), which considers a sequence of tasks, each defining a Markov Decision Process (MDP) $M_t = < S_t, A_t, T_t, r_t, \gamma >$ such that $S$ is the state space, $A$ is the action space, $p : S \times A \to \Delta S$ is the transition probability where $\Delta(S)$ is the probability simplex over $S$, $r : S \times A \to \mathcal{R}$ is the reward function so $r_t(s_t, h, a_{t,h})$ is the immediate reward in task $t$ when taking action $a_{t,h}$ at state $s_{t,h}$, $h$ indexes the environment step, and $\gamma \in [0, 1)$ is the discounted factor. Continual RL aims to achieve a policy $\pi_\theta$ at task $T$ that performs well (with high expected return) on all seen tasks $t \leq T$ , with only a limited (or without) buffer of previous tasks' experiences:

$$\theta^* = \underset{\theta}{argmax} \sum_{t=1}^{T} \mathcal{E}_{\pi_\theta}[\sum_{h=0}^{\infty} \gamma^h r_t(s_{t,h}, a_t, h)] \tag{1}$$

Continual learning is a natural human skill that can accumulate knowledge generalizable to new tasks without forgetting the learned ones. However, RL agents often struggle with achieving the goal in Eq. 1 due to the plasticity-stability trade-off: the policy is expected to quickly adapt to new tasks $t \geq T$ (plasticity) but meanwhile to retain its performance on previous tasks $t < T$ (stability).

Existing strategies for continual RL mainly focus on improving stability and reducing catastrophic forgetting. Rehearsal-based methods such as CLEAR(Rolnick et al., 2019) and PC (Schwarz et al., 2018) repeatedly replay buffed experiences from previous tasks but the required buffer memory and computational cost grow with the number of tasks(Kumari et al., 2022). By contrast, Regularization-based methods such as EWC(Kirkpatrick et al., 2017) and PC(Kaplanis et al., 2019) alleviate forgetting without the replay buffer by adding extra regularizers when learning new tasks, which can bias the policy optimization and lead to sub-optimal solutions(Zhao et al., 2023). Finally, structure-based methods adopt separate modules, i.e., sub-networks within a fixed-capacity policy network, for each task(Mendez & Eaton, 2022). We summarize two main categories of structure-based methods in the following.

**Connection-level methods.** This category includes methods such as PackNet(Mirchandani et al., 2021), Sup-Sup(Wortsman et al., 2020), and WSN(Kang et al., 2022). For task $t$, the action at is drawn from $a_t \sim \pi(s_t; \theta \otimes \phi_t)$ where $s_t$ is the state and $\phi_t$ is a binary mask applied to the model weights $\theta$ in an element-wise manner (i.e., $\otimes$). Pack-Net generates $\phi$ t by iteratively pruning $\theta$ after the learning of each task, thereby preserving important weights for the task while leaving others for future tasks. SupSup fixes a randomly initialized network and finds the optimal $\phi_t$ for each task $t$. WSN jointly learns $\theta$ and $\phi_t$ and uses Huffman coding (Huffman, 1952) to compress $\phi_t$ for a sublinear growing size of $\{\phi_t\}_{t=1}^{T}$ with increasing tasks. However, these methods usually need to store the task-specific masks for each task in history, leading to additional memory costs(Huang et al., 2022). Moreover, their masks are seldom optimized for knowledge sharing across tasks, impeding the learned policy from being generalized to unseen tasks.

**Neuron-level methods.** Instead of extracting task-specific sub-networks by applying masks to model weights, the other category of methods produces sub-networks by applying masks to each

layer's neurons/outputs of a policy network. Compared to connection-level methods, they use layer-wise masking to achieve a more flexible and compact representation of sub-networks. But the generation of masks depends on either heuristic rules or computationally inefficient policy gradient methods. By contrast, LLIT generates masks by highly efficient sparse coding (solving a relatively small lasso problem).

## 3 METHOD

To facilitate more efficient execution of CRL tasks in a task-agnostic setting, as well as to mitigate the occurrence of catastrophic forgetting and promote knowledge transfer, we propose our own approach by leveraging the general inferential capabilities of large language models and the fine-tuning capabilities of the Decision Transformer model.

### 3.1 ROUGH INFERENCE WITH LARGE LANGUAGE MODEL

In the context of Continual Reinforcement Learning (CRL), an agent is tasked with performing a continuous sequence of multiple tasks. During the learning process, the identification of tasks (referred to as Task IDs) and task boundaries (Task Boundary) are critical for strategies employed by certain CRL methods. These task identifiers are often used for switching training datasets, storing data in different replay buffers, or altering policy distributions. However, in task-agnostic settings and real-world scenarios, agents often struggle to directly obtain accurate Task IDs or task boundary indications during training.

Hence, the goal of our proposed approach is to efficiently guide the agent's learning without relying on human's intervention. We recognize that although agents cannot obtain task-specific information solely from themselves, they do have direct access to observations, such as sensor data, and executable actions, such as robotic arm movements. The semantics of various dimensions of this observational and actionable data are well-defined for agents.

Assume the agent is trained on the task $t$ in a task sequence which contains $\mathcal{T}$ tasks in total, and the observation space and action space of it are denoted as $\mathcal{O}_t$ with $n$ dimensions, and $\mathcal{A}_t$ with m dimensions separately. The simple language descriptions about different dimensions of $\mathcal{O}_t$ and $\mathcal{A}_t$ can be collected as two sets: $Des_{\mathcal{O}_t} = \{des^i_{\mathcal{O}_t}\}^n_{i=1}$ and $Des_{\mathcal{A}_t} = \{des^j_{\mathcal{A}_t}\}^m_{j=1}$, where $des^i_{\mathcal{O}_t}$ represents the description of $i_{th}$ dimension of observation space $\mathcal{O}_t$ and $des^j_{\mathcal{A}_t}$ represents the description of $j_{th}$ dimension of action space $\mathcal{A}_t$. Large language models possess fundamental reasoning ability based on general knowledge, which implies that they can roughly infer task-related information when they receive given description sets about observation space and action space. In LLIT, we utilize LLMs to focus on inferring two types of crucial task-related information:**task content** and **task tips**.

Task content, denoted as $l^t_{content}$, should be a short sentence that briefly summarizes the task, and can be viewed as a prediction via LLMs about task name, while task tips, denoted as $l^t_{tips}$, are a set of suggestions in natural language, provided by LLMs on the purpose of instructing the agent to accomplish the task more efficiently. To acquire more precise content and meaningful tips of a task, we carefully design prompt templates that are input into LLMs with the description sets of the task. This process can be written as:

$$l^t_{content} = f_{LLM}(Des_{\mathcal{O}_t}, Des_{\mathcal{A}_t}; p_{content})$$
$$l^t_{tips} = f_{LLM}(Des_{\mathcal{O}_t}, Des_{\mathcal{A}_t}, l^t_{content}; p_{tips}) \tag{2}$$

where $f_{LLM}$ denotes a LLM function, $p_{content}$ denotes the prompt template for inferring the task content of task $t$ and $p_{tips}$ denotes the prompt template for proving the task tips for task $t$. Extracting task content and task tips through LLMs offers three distinct advantages. Firstly, $l^t_{content}$ and $l^t_{tips}$ represent high-level semantic understandings of the original information, encapsulating human knowledge about the task in the form of natural language, which means they can serve as signals besides rewards for guiding the optimization of policy. Secondly, $l^t_{content}$ and $l^t_{tips}$ are representations closely associated with task-specific information, facilitating knowledge transfer across different tasks. Lastly, obtaining $l^t_{content}$ and $l^t_{tips}$ only requires initial acquisition at the beginning of training for each task, leading to less computation.

## 3.2 GROUNDING TIPS TO POLICY

One of the expected effects of our proposed framework is that task tips containing human knowledge can be used to guide agent learning, thereby improving efficiency and interpretability. However, there are two challenges to deploy the task tips in the agent's policy. Firstly, there is a huge gap between natural language(the form of tips) and sensor data(the form of observation), which suggests that it is necessary to transform the tips into the observation space. Secondly, it is difficult to translate the knowledge in the task tips instead of text itself into effective reward signals.

To address the challenges above, we train a reward model to bridge the tips and observation space. Firstly, each of the tips can be parsed by a frozen similarity model.We utilize the similarity model to detect if the description of a dimension of observation space exists in a tip, while those existed will be bound with their relevant sub-sentence in the tips and others will be bound with a predefined character "¡NULL¿".Therefore, each tip will be parsed in a vector that has the same dimensions as the observation space, for example, the $i_{th}$ parsed tip could be $D_t^i = \{< ob\_1\_sub\_tip >, \ldots, < NULL >, \ldots \}$.

When we obtain the parsed tips of a task, we establish a correspondence between the observation space and tips, which helps generate the auxiliary reward signal to instruct the policy to update. Specifically, we select a pre-trained tokenizer to turn each parsed tip and each single observation data into token embeddings denoted as $e_{tip}$ and $e_o$ separately, then element-wise concatenate $e_{tip}$ and $e_o$. Finally, the concatenated embedding will be the input to train a transformer model, which is an auxiliary reward model.The process can be shown as:

$$R_a = f_{ARM}([e_{tip}; e_o]) \tag{3}$$

where $R_a$ is auxiliary reward, $f_{ARM}$ is the auxiliary reward model which based on transformer, and $[e_{tip}; e_o]$ is the element-wise concatenation of $e_{tip}$ and $e_o$.

## 3.3 LEARNING WITH POOL OF INFERRED INFORMATION

After sufficient training of the auxiliary reward model, we obtain a tool to extract the semantics of tips from human knowledge and instruct the agent to learn more efficiently and reasonably, and next we need to solve the problem that the nature of continual reinforcement learning brings. In CRL, agent needs to learn tasks sequentially with single policy model, which lead to catastrophic forgetting if doing nothing to the intervene between tasks.

To deal with catastrophic forgetting, we propose a modulation pool. Similar to L2P,we define a modulation pool that contains a set of M keys,$K_{pool} = k_i \mid_{i=1}^M$.Each $k_i$ is associated with a set of modulation vectors$l_{k,b}^i, l_{v,b}^i, l_{ff,b}^i$ as values,for each layer block b of a DT with $B$ layer blocks,where $l_k \in \mathcal{R}^{d_k}, l_v \in \mathcal{R}^{d_v}$,and $l_{ff} \in \mathcal{R}^{d_{ff}}$, $d_k, d_v$,and $d_{ff}$ correspond to the dimensions of the keys, queries, and feedforward activations in the DT, respectively. Since we follow a GPT-2-like architecture, $d_k = d_v$ and $d_{ff} = 4 \times d_k$. We interleave each Transformer layer with separate modulation vectors, resulting in $d_k + d_v + 4 \times d_{ff}$ learnable parameters per layer. At time $t$,we compose all states in a trajectory $\tau$ into a matrix $S_{\leq t}$ after the y are processed via the embedding layer of the DT. Subsequently, we reduce the matrix to a query vector $q_t \in R^{d_q}$ by an aggregation function $g(\cdot)$:

$$q_t = g(\mathcal{S}_{\leq t}) \tag{4}$$

For the aggregation function $g(\cdot)$, we use mean-pooling by default. Further, we retrieve a set of modulation vectors $\{l_{k,b}^j, l_{v,b}^j, l_{ff,b}^j\} \mid_{b=1}^B$ by the maximum similarity between each $k \in K_{pool}$ in the modulation pool and the query $q_t$ at timestep $t$:

$$j = \underset{k \in K_p}{argmax} sim(q_t, k)n(k)^{-1} \tag{5}$$

In our case, $sim(\cdot, \cdot)$ corresponds to the cosine similarity and $n(k)^{-1}$ represents the inverse selection count for key k up to the current task. This way, we discourage that queries for different tasks attend to the same key. Subsequently, we use $\{l_{k,b}^j, l_{v,b}^j, l_{ff,b}^j\} \mid_{b=1}^B$ to modulate the attention mechanism in the DT, as proposed by Liu et al.(2022):

$$(l_v^j \odot V)^T softmax(\beta(l_k^j \odot K)Q) \tag{6}$$

Here, $\odot$ corresponds to element-wise multiplication, and $\beta = \frac{1}{\sqrt{d_k}}$. Also, $Q$, $K$, $V$ refer to queries, keys, and values in the self-attention, respectively. Further, $l_{ff}^j$ modulates the activations of the position-wise feed-forward activations in DT. All modulation vectors are initialized to ones, and, thus, the activations remain unchanged at the start of training. All keys in $K_{pool}$ are initialized uniformly between [-1,1]. Our method unifies the benefits of both, (IA)3 and L2P in the RL setting. It assures high-performance and few additional learnable parameters, while it avoids forgetting on the pre-trained tasks. Moreover, it provides a simple task-matching mechanism and enables scalability to numerous tasks.

## 4 EXPERIMENTS SETUP

### 4.1 ENVIRONMENTS AND TASKS

In order to assess the CRL ability of LLIT and the baselines in a cross-domain, task-agnostic setting, we focus on environments with significant differences in domains, specifically where the state space and action space, vary. Each environment consists of multiple control tasks. We randomly select tasks from different environments in proportion and shuffle them to create a mixed control task sequence. In this setup, the task sequence allows for a comprehensive evaluation of LLIT and baselines' knowledge transfer capabilities and generalization performance when dealing with tasks that exhibit significant differences in control objects, control logic, task goals, and other aspects. We list the evaluation environments below:

**Classical Control**:The Classical Control environment consists of 2D control tasks from OpenAI Gym, including Cart Pole, Inverted Pendulum, LunarLander, and others. In these tasks, the controlled object remains within a 2D plane, and their control physics are relatively straightforward.

**Mujoco Control**:The Mujoco Control environment consists of multiple control tasks with Mujoco physics engine.These tasks are selected from two benchmarks: OpenAI Gym Mujoco and the DM-control benchmark.In mujoco control tasks, The controlled objects typically have multiple joints, and there are simulated physical interactions between these joints.

**Continual World**:The Continual World environment is a task sequence originated from the Continual World benchmark, which contains 20 realistic robotic tasks carefully selected and arranged from the Meta World benchmark.These tasks and their ordering are arranged based on their transfer matrix so that there is a high variation of forward transfers.In our setup, the Continual World environment will be an independent evaluated environment where the tasks will maintain their original arrangement instead of being randomly inserted into the mixed task sequence.

### 4.2 BASELINES

We compare LLIT with several baselines and state-of-the-art (SoTA) continual RL methods. According to (Lange et al., 2022), these methods can be divided into three categories: regularization-based, structure-based, and rehearsal-based methods. Concretely, regularization-based methods include L2, Elastic Weight Consolidation (EWC) (Kirkpatrick et al., 2017), Memory-Aware Synapses (MAS) (Aljundi et al., 2018), and Variational Continual Learning (VCL) (Nguyen et al., 2018). Structure-based methods include PackNet (Mallya and Lazebnik, 2018), Hard Attention to Tasks (HAT) (Serr'a et al.,2018), and TaDeLL (Rostami et al., 2020). Rehearsal-based methods include Reservoir, Average Gradient Episodic Memory (A-GEM) (Chaudhry et al., 2019), and ClonEx-SAC (Wolczyk et al., 2022). For completeness, we also include a naive sequential training method (i.e., Finetuning) and representative multi-task RL baselines (MTL (Yu et al.,2019) and MTL+PopArt (Hessel et al., 2019)), which are usually regarded as the soft upper bound a continual RL method can achieve. For a fair comparison, we refer to the Continual World repository for implementation and hyper-parameter selection. We re-run these methods to ensure the best possible performance. In addition, we adopt author-reported results for ClonEx-SAC due to the lack of open-sourced implementation.

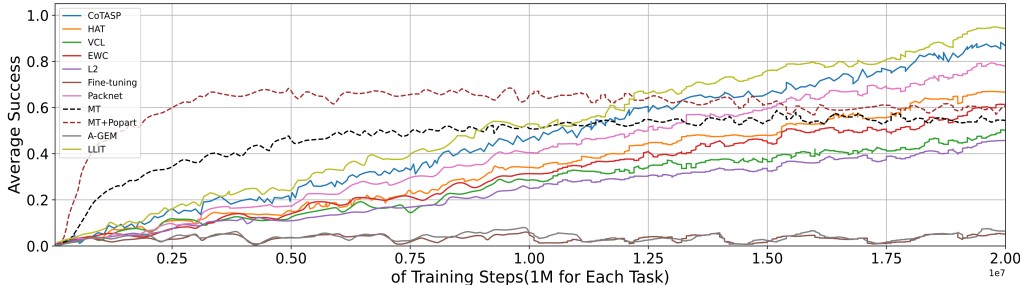

Figure 2: **Performance ($mean \pm std$ over 5 random seeds) of all methods on CW20 sequence**. LLIT outperforms all the continual RL methods and all the multi-task RL baselines.

| Benchmarks | | CW10 | | | CW20 | | |
|---|---|---|---|---|---|---|---|
| **Metrics** | | $P \uparrow$ | $F \downarrow$ | $G \downarrow$ | $P \uparrow$ | $F \downarrow$ | $G \downarrow$ |
| CL | $L2$ | $0.41 \pm 0.14$ | $0.00 \pm 0.03$ | $0.53 \pm 0.05$ | $0.51 \pm 0.07$ | **-0.09** $\pm 0.04$ | $0.56 \pm 0.06$ |
| | $EWC$ | $0.62 \pm 0.15$ | $0.01 \pm 0.04$ | $0.36 \pm 0.05$ | $0.60 \pm 0.08$ | $0.03 \pm 0.03$ | $0.40 \pm 0.07$ |
| | $MAS$ | $0.63 \pm 0.17$ | **-0.04** $\pm 0.03$ | $0.45 \pm 0.05$ | $0.49 \pm 0.04$ | $0.02 \pm 0.02$ | $0.51 \pm 0.02$ |
| | $VCL$ | $0.49 \pm 0.09$ | $-0.01 \pm 0.07$ | $0.44 \pm 0.05$ | $0.52 \pm 0.13$ | $-0.02 \pm 0.06$ | $0.53 \pm 0.05$ |
| | $Finetuning$ | $0.11 \pm 0.03$ | $0.72 \pm 0.03$ | $0.25 \pm 0.07$ | $0.04 \pm 0.00$ | $0.74 \pm 0.04$ | $0.32 \pm 0.04$ |
| | $PackNet$ | $0.82 \pm 0.11$ | $0.00 \pm 0.06$ | $0.25 \pm 0.06$ | $0.76 \pm 0.06$ | $0.00 \pm 0.00$ | $0.31 \pm 0.04$ |
| | $HAT$ | $0.64 \pm 0.11$ | $0.00 \pm 0.00$ | $0.45 \pm 0.06$ | $0.64 \pm 0.07$ | $0.00 \pm 0.00$ | $0.45 \pm 0.04$ |
| | $A-GEM$ | $0.12 \pm 0.05$ | $0.65 \pm 0.03$ | **0.23** $\pm 0.02$ | $0.06 \pm 0.02$ | $0.71 \pm 0.07$ | $0.27 \pm 0.04$ |
| | $ClonEx-SAC*$ | $0.86$ | $0.02$ | $-$ | $0.87$ | $0.02$ | $-$ |
| | $CoTASP$ | $0.91 \pm 0.03$ | $0.00 \pm 0.00$ | $0.25 \pm 0.03$ | $0.86 \pm 0.02$ | $0.00 \pm 0.00$ | **0.25** $\pm 0.03$ |
| MT | $MTL$ | $0.51 \pm 0.10$ | $-$ | $-$ | $0.50 \pm 0.12$ | $-$ | $-$ |
| | $MTL+PopArt$ | $0.71 \pm 0.13$ | $-$ | $-$ | $0.67 \pm 0.16$ | $-$ | $-$ |
| | $LLIT(ours)$ | **0.95** $\pm 0.07$ | $0.00 \pm 0.00$ | $0.19 \pm 0.06$ | **0.91** $\pm 0.04$ | $0.00 \pm 0.00$ | **0.27** $\pm 0.03$ |

Table 1: **Evaluation ($mean \pm std$ of 3 metrics over 5 random seeds) on Continual World**. *-reported in previous work. Reg = Regularization-based, Struc = Structure-based, Reh = Rehearsal-based, MT = Multi-task, $P$ = Average Performance, $F$ = Forgetting, $G$ = Generalization. The best result for each metric is highlighted.

## 4.3 EVALUATION

Following a widely-used evaluation protocol in continual learning literature, we adopt three metrics. (1) Average Performance (higher is better): the average performance at time $t$ is defined as $P(t) = \frac{1}{T} \sum_{i=1}^{T} p_i(t) pi(t)$ where $p_i(t) \in [0, 1]$ denotes the success rate of task i at time $t$. This is a canonical metric used in the continual learning community. (2) Forgetting (lower is better): it measures the average degradation across all tasks at the end of learning, denoted by $F = \frac{1}{T} \sum_{i=1}^{T} p_i(i \cdot \delta) - p_i(T \cdot \delta)$, where $\delta$ is the allowed environment steps for each task. (3) Generalization (lower is better): it equals to the average number of steps needed to reach a success threshold across all tasks. Note that we stop the training when the success rate in two consecutive evaluations reaches the threshold (set to 0.9). Moreover, the metric is divided by $\delta$ to normalize its scale to $[0, 1]$.

## 5 EXPERIMENTS

### 5.1 CONTINUAL LEARNING EXPERIMENTS

This section presents the comparison between LLIT and ten representative continual RL methods on mixed task sequence and CW benchmarks. We focus on the stability (retain performance on seen tasks) and the plasticity (quickly adapt to unseen tasks) and keep the constraints on computation, memory, number of samples, and neural network architecture constant. Table 1 summarizes our main results on CW10 and CW20 sequences. LLIT consistently outperforms all the compared methods across different lengths of task sequences, in terms of both average performance (measures stability) and generalization (measures plasticity). We observe that when the hidden-layer size is

| Benchmark | CW20 | |
| --- | --- | --- |
| Metrics | $P(\uparrow)$ | $G(\downarrow)$ |
| LLIT(ours) | $0.91 \pm 0.02$ | $0.27 \pm 0.03$ |
| with $D$ frozen | $0.73 \pm 0.06$ | $0.47 \pm 0.03$ |
| with $\alpha$ frozen | $0.79 \pm 0.06$ | $0.34 \pm 0.02$ |
| with both frozen | $0.62 \pm 0.05$ | $0.52 \pm 0.03$ |
| lazily update $D$ | $0.85 \pm 0.03$ | $0.29 \pm 0.05$ |
| EWC | $0.60 \pm 0.07$ | $0.39 \pm 0.06$ |
| PackNet | $0.78 \pm 0.07$ | $0.32 \pm 0.04$ |
| A-GEM | $0.08 \pm 0.02$ | $0.29 \pm 0.04$ |
| Finetuning | $0.05 \pm 0.00$ | $0.30 \pm 0.05$ |

Table 2: **Ablation study.** Performance of LLIT variants on CW20 sequence. Please refer to Sec. 4.3 for a detailed explanation.

the same as other structure-based methods (PackNet and HAT), LLIT outperforms them by a large margin, especially in the generalization metric, indicating the advantage of LLIT in improving the adaptation to new tasks.

Moreover, we find that most continual RL methods fail to achieve positive backward transfer (i.e., $F < 0$) except for VCL, suggesting the ability to improve previous tasks' performance by learning new ones is still a significant challenge. We leave this for future work. Finally, the results in Fig. 3 show that LLIT is the better method than CoTASP that performing comparably to the multi-task learning baselines on the first ten tasks of CW20 sequence, and it exhibits superior performance over these baselines after learning the entire CW20 sequence. One possible explanation is that the knowledge accumulated by LLIT's meta-policy network and dictionaries leads to improved generalization.

**Effectiveness of core designs.** To show the effectiveness of each of our components, we conduct an ablation study on four variants of LLIT, each of which removes or changes a single design choice made in the original LLIT. Table 2 presents the results of the ablation study on CW20 sequence, using two representative evaluation metrics. Among the four variants of LLIT, "D frozen" replaces the learnable dictionary with a fixed, randomly initialized one; "$\alpha$ frozen" removes the prompt optimization proposed in Sec. 3.3; "both frozen" neither updates the dictionary nor optimizes the prompt; "lazily update D" stops the dictionary learning after completing the first ten tasks of CW20 sequence. According to the results in Table 2, we give the following conclusions: (1) The use of a fixed, randomly initialized dictionary degrades the performance of LLIT on two evaluation metrics, highlighting the importance of the learnable dictionary in capturing semantic correlations among tasks. (2) The "$\alpha$ frozen" variant performs comparably to our LLIT but outperforms the results achieved by EWC and PackNet. This indicates that optimizing the prompt can improve LLIT's performance but is not crucial to our appealing results. (3) The "both frozen" variant exhibits noticeable degradation in performance, supporting the conclusion that the combination of core designs proposed in LLIT is essential for achieving strong results. (4) The "lazily update D" variant only slightly degrades from the original LLIT on the performance but still outperforms all baselines by a large margin, indicating that the learned dictionary has accumulated sufficient knowledge in the first ten tasks so that LLIT can achieve competitive results without updating the dictionary for repetitive tasks.

**Effect of key hyperparameters.**LLIT introduces the sparsity parameter $\lambda$, a hyperparameter that controls the trade-off between the used network capacity and the performance of the resulting policy. A larger value of $\lambda$ results in a more sparse policy sub-network, improving the usage efficiency of the meta-policy network's capacity. But the cost is decreased performance on each task due to the loss of expressivity of the over-sparse task policy. According to the results in Fig. 3, LLIT with $\lambda$=1e-3 or 1e-4 achieves better trade-off between performance and usage efficiency than other structure-based methods (HAT and PackNet) on CW10 sequence.

## 6 CONCLUSION

We propose LLIT to address two key challenges in continual RL: (1) training an auxiliary reward model using language instructions interpretable and generalizable to all seen and even unseen tasks; (2) efficiently extracting similarities in semantics of tasks and mitigating forgetting. LLIT learns a policy with embeddings of human skills in the form of language and a prompt pool to transfer knowledge across tasks. This encourages knowledge sharing/reusing among relevant tasks while reducing harmful cross-task interference that causes forgetting and poor new task adaptation. Without any experience replay, LLIT achieves a significantly better plasticity-stability trade-off and more efficient network capacity allocation than baselines. Its extracted policies outperform all baselines on both previous and new tasks.

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

## A  APPENDIX

