# OpenReview forum: "Learning with Language Inference and Tips for Continual Reinforcement Learning"
_ICLR.cc/2024/Conference — ICLR 2024 Conference Withdrawn Submission_

### Official Review · Reviewer_LukW · 2023-10-29

**Soundness:** 1 poor
**Presentation:** 1 poor
**Contribution:** 2 fair
**Rating:** 1
**Confidence:** 5

**Summary:**

This paper presents “Learning with Language Inference and Tips (LLIT)”. It introduces a rewarding mechanism to parse and ground human knowledge in natural language form to the task space and produces an interpretable policy for each task in a task-agnostic setting.

The **challenge** it solves involves the need for the agent to adapt quickly to new tasks while retaining knowledge from previous tasks.

LLIT promotes the sharing and reuse of knowledge among tasks that are closely related, all while mitigating the harmful effects of cross-task interference, which often leads to forgetting and poor adaptation to new tasks.

**Strengths:**

1. This paper presents an interesting and novel way to adapt to unseen tasks for continual RL with the help of LLM.

2. Authors compare LLIT with several baselines and state-of-the-art (SoTA) continual RL methods.

**Weaknesses:**

1. This paper is **incomplete**.
2. They mentioned many tasks in the experiments setup section but did NOT run any experiments on them, such as Mujoco Control and Classical Control.
3. **No details** about the whole architecture. No hyper-parameter or structure details are reported at all.
4. The results in Figure 2 is not complete. We can see most of the baselines and models **did not converge**.
5. No parameters/ time complexity reported. This architecture requires much more parameters and computing resources.

**Questions:**

Pls, make sure all the references are correct. ("while the learned prompt pool captures the semantic correlations among tasks(Fig.??)".)

See the weakness. I recommend the authors to withdraw and finish the paper first.

---

### Official Review · Reviewer_8SCJ · 2023-11-02

**Soundness:** 1 poor
**Presentation:** 1 poor
**Contribution:** 2 fair
**Rating:** 3
**Confidence:** 3

**Summary:**

This paper proposes a continual RL technique which leverages a pretrained LLM to produce task descriptions (based on action + observation specs), which are used to train an auxiliary reward model as well as keep a dictionary of task-specific embedding.

They demonstrate its performance on the Continual World benchmark (which automates the data collection, policy learning and task curriculum aspect, while also providing most of their baselines), and show that their method outperforms all other methods.

Overall, this was an interesting paper, but in this current state it lacks too many details to understand exactly what was done and how many of their choices were effectively implemented in CW. It would need quite a lot of improvement to get up to ICLR’s standard in time for publication.

**Strengths:**

1. The paper leverages LLMs in a novel way for continual RL.
2. Results do appear strong, so if CW is accepted as a benchmark this may be important to a subset of the community. I was not aware of CW till now.

**Weaknesses:**

1. There is a lack of details throughout the paper, with decisions not being fully explained, many missing examples of parts of the process, and finally no Appendix to give any details for how most of this was parametrised and implemented. In this state this is not reproducible research.
2. Several aspects of the model are not introduced well enough. Even though a lot of space is spent talking about how the LLM is used, and why using one is useful, I do not think I understand the details of how all components fit together still.
3. Key parameters are not defined (e.g. D, alpha, lambda) and there is no Appendix to help.
4. Overall the paper feels rushed and would need some iterations to reach ICLR’s expected threshold.

**Questions:**

1. What does CW provide, and what exactly are you modifying in it?
   1. I had to look at their paper to see that it automates the data collection + policy learning aspect, there is no mention of any of that in the paper, to the point where I thought you were just using some offline RL data that was pre-collected?
   2. How is the Auxiliary Reward model used? Where is it connected and how does it influence the behaviour?
   3. How is the similarity reward used?
   4. What parts of these are functions of time/observation or just constant per task?
2. There are no details about the LLM behaviour and outputs.
   1. What are the prompts used?
   2. Why did you split them between task tips and task content?
   3. What does the model output? You need to provide examples on a few tasks.
   4. How do you obtain the observation and action specs? How would that inform how to solve the task?
3. There are not enough details about the Grounding Tips to Policy section:
   1. Why did you have to map the tips to the observation specs in the manner described in 3.2? What happens if you did not do that? Did you try other methods?
   2. How is f_ARM trained? Is it trained? On what data? To predict what target?
4. Section 3.3 contains many low-level details yet is not clear enough about what it presents.
   1. It uses entirely different notation and semantics than what is in Figure 1. It is the “Modular Pool Model”, correct?
   2. Where is e_tn?
   3. How are these trained? Which data, etc?
5. Again Section 4 contains no details at all, or assumes full knowledge of what Continual World provides. This is not acceptable IMO.
6. Section 5.1 uses different notation yet again and isn’t clear enough
   1. What is alpha? D? lambda?
   2. Are these the most important parameters/aspects to modify?
   3. What about the way to prompt the LLM?

---

### Official Review · Reviewer_QZUw · 2023-11-04

**Soundness:** 2 fair
**Presentation:** 2 fair
**Contribution:** 2 fair
**Rating:** 3
**Confidence:** 4

**Summary:**

The paper proposes a method for continual RL that leverages LLMs to infer task-related information and tips in natural language. The proposed method, LLIT, uses the inferred tips to train an auxiliary reward model that guides the policy optimization. Besides, LLIT uses a prompt pool to capture semantic correlations among tasks and extract policy sub-networks for each task. The paper evaluates LLIT on several CRL benchmarks and shows that it outperforms existing methods in terms of stability, plasticity, and generalization.

**Strengths:**

The paper introduces an effective way to use LLMs for CRL. The experiments and ablation studies to demonstrate the advantages of LLIT over baselines.

I think the research topic is interesting, and CRL is an important area to be explored. It seems that LLMs can provide some guidance as it contains rich prior knowledge.

**Weaknesses:**

Though I think the paper is addressing an interesting question, there are many typos in the paper, making it seem like an incomplete version.

Besides, the motivation/objective of auxiliary reward are unclear. The similarity model is used for measuring the semantic similarity, and can not be used to provide reward signal. In this way, the auxiliary reward is not effective for policy learning.

Overall, there lacks some figures (e.g., Effect of key hyperparameters, Fig 3), and some important information that support the proposed method. I think the current version does not reach the acceptance line of ICLR. However, I do hope authors can provide more information in the discussion period.

**Questions:**

It is unclear how is the auxiliary reward model trained. Section 3.2 first mentions that auxiliary reward signal can be generated from parsed tips. But it also says there is an auxiliary reward model.

---

### Meta-Review · Area_Chair_PFLB · 2023-12-05

**Metareview:**

All reviewers found serious issues with the completeness of the paper.

**Justification For Why Not Higher Score:**

All reviewers found serious issues. There's no author response.

**Justification For Why Not Lower Score:**

N/A

---

### Decision · Program_Chairs · 2024-01-16

Reject